# Vegetation Cover Management and Landscape Plant Species Composition Influence the Chrysopidae Community in the Olive Agroecosystem

**DOI:** 10.3390/plants11233255

**Published:** 2022-11-27

**Authors:** Rafael Alcalá Herrera, Antonio García-Fuentes, María Eugenia Ramos-Font, Mª Luisa Fernández-Sierra, Francisca Ruano

**Affiliations:** 1Department of Agronomy, University of Córdoba, Campus de Rabanales, Building C4 “Celestino Mutis”, 14014 Córdoba, Spain; 2Department of Environmental Protection, Estación Experimental del Zaidín (EEZ-CSIC), C/Profesor Albareda 1, 18008 Granada, Spain; 3Departamento de Biología Animal, Biología Vegetal y Ecología, Edificio B3, Universidad de Jaén, Campus Las Lagunillas s/n, 23071 Jaén, Spain; 4Servicio de Evaluación, Restauración y Protección de Agrosistemas Mediterráneos, Estación Experimental del Zaidín (EEZ-CSIC), C/Profesor Albareda 1, 18008 Granada, Spain; 5Department of Zoology, University of Granada, Campus de Fuentenueva s/n, 18071 Granada, Spain

**Keywords:** ecological infrastructures, cover crops, patch vegetation, *Olea europaea*, *Chrysoperla*, *Apertochrysa*, *Pseudomallada*

## Abstract

Habitat manipulation through the promotion of semi-natural habitats such as cover and patch vegetation is a possible means of offsetting the negative impacts of the agricultural practices. A baseline situation is crucial before any successful habitat manipulation is attempted. We studied the effects that current vegetation cover management practices have on plant composition and the potential attraction that the plant families from the semi-natural habitats could have on the Chrysopidae community, a key pest control agent, in five olive farms in Granada (Spain). Vegetation cover was assessed using a point quadrat methodology in eight transects per farm. In addition, the patch vegetation was characterized with 60 transects using a line intercept methodology. The woody patch vegetation and olive tree canopies were vacuumed using a field aspirator to collect adult Chrysopidae. In the cover vegetation we observed great variability in both the richness and diversity of plant communities caused by the vegetation cover management techniques and the transect position (in the middle of the rows or beneath the tree canopy). The plant families with the greatest plant cover were the Asteraceae and Fabaceae, where Asteraceae was favoured by tillage and Fabaceae by grazing, while in the patch vegetation, the predominant families were the Rosaceae and Fagaceae. Our results indicate that the genus *Chrysoperla* was mostly correlated with the Plantaginaceae, Brassicaceae and Asteraceae plant families in the cover vegetation, and with the Caryophyllaceae and Rosaceae families in the patch vegetation. The genera *Apertochrysa* and *Pseudomallada* were associated with the families Malvaceae and Poaceae in the cover vegetation, and with the families Cupressaceae, Poaceae and Pinaceae in the patch vegetation. Our study shows to the farmers the possibilities of vegetation cover management to select plant families for the cover vegetation.

## 1. Introduction

One of the challenges facing agricultural systems today is how to improve sustainability and reduce dependence on external inputs [1]. Restoring ecosystem services such as natural pest control, pollination, and soil nutrient recycling, fertility and structure, are among the priorities when aiming to move towards sustainable agriculture [2]. One of the ten objectives of the new European Common Agricultural Policy is to contribute to the conservation or restoration of habitats or species to protect of biodiversity, as well as the maintenance and creation of landscape features or non-productive areas to enhance ecosystem services, with direct (eco-schemes) and conditionally payments [3]. One solution is to promote semi-natural habitat (SNH) such as cover and/or patch vegetation [4,5,6], which is defined as vegetation that grows within or outside the crop areas that protects soils during the productive period [7].

Cover vegetation has multiple benefits for soils as it increases organic carbon, reduces water runoff and soil loss and compaction, and lowers soil temperatures [5,8]. As well, cover vegetation improves the structure and water properties of the soil and enhances nutrient recycling and microbial activity [5,9]. Furthermore, cover and patch vegetation provides food resources, refuge and reproduction sites for the natural enemies (predators and parasitoids) that play a key role in pest control in crops and, specifically, in olive farms [8,10,11,12]. Given their key role in crop protection as well as their presence in crops, one of the most studied group of predators in olive agroecosystems are the family Chrysopidae [13,14,15]. Chrysopidae constitute an efficient biological control agent in the predation of the olive moth, *Prays oleae* (Bernard, 1788), one of the principal olive pests [13,16,17,18]. The integrated pest management guide recommends the use of Chrysopidae as one biological control strategy against *P. oleae* infestation in olive orchards [19]. Furthermore, Chrysopidae can be found almost all year round in the olive agroecosystem; they are dependent on the surrounding vegetation [11,20,21,22]. Previous studies have reported that there is an association between the Chrysopidae species and the types of vegetation [18,20,23]. Although the diet of mostly adult Chrysopidae species is palyno-glycophagous, depending on the vegetation to feed on nectar, pollen and honeydew [24], their larval stages prey on a wide range of soft-bodied arthropods such as aphids, whiteflies, psyllids, thrips, psocids, moths, mites and leaf-hoppers [25,26].

Olive trees (*Olea europaea* L.) were first cultivated around 6000 years ago in the Middle East, from where they have spread throughout the Mediterranean Basin [27]. Today, oliviculture is performed in 58 countries on five continents. Olive farms cover 11.5 million ha of land worldwide [28] and Spain—above all, Andalusia—is a world leader in terms of the surface area devoted to olive farms (1.6 million ha) [29]. The expansion and intensification of agricultural practices, which has also affected olive farms, have led to landscape simplification and a reduction in SNH, and caused a loss of biodiversity [30,31]. Vegetation cover management in olive farms mostly consists of tillage and/or herbicide application to minimize competition for water resources and nutrients between the cover vegetation and olive trees [32,33]. Although the combination of intense precipitation events and bare soil can cause nutrient and soil loss, olive farmers generally do not perceive any short-term yield reduction as a result due to the benefits they derive from mechanical management techniques and the provision of agrochemical inputs [33,34,35].

Previous studies have concluded that cover vegetation in olive farms, both temporary and permanent, sown and spontaneous, reduces soil loss, controls soil erosion and improves the properties of the soil [8,33,36,37]. Other studies have focused on determining the composition of the plant species that appear in the cover vegetation [8,35,38,39,40]. Gómez et al. [8] point out that the ways of developing suitable cover vegetation will vary from farm to farm. Cover vegetation is a tool that needs time, knowledge and economic resources if it is to be implemented successfully, and may even be viewed with suspicion by some farmers due to water shortages, especially in Mediterranean agroecosystems characterized by semi-arid climatic conditions [40]. However, the surface area of cover vegetation in olive farms in Spain has increased by 181,515 ha over the past 10 years; in Andalusia, vegetation cover management in farms is now closely split between areas under tillage (741,705 ha) and areas with cover vegetation (703,626 ha) [41]. The Andalusian patch vegetation is composed mainly of the remnants natural to the meso-Mediterranean holm oak *Paeonio-Querco rotundifoliae sigmetum* vegetation series located in the mountainous massifs and on the steepest hills, whose slope and lithology have prevented the colonisation of olive trees [42,43,44]. In this context and as a prior step to successful establishment of a cover vegetation (sown or spontaneous) or landscape features, it is vital to determine the spontaneous plant species in SNH and how they are affected by vegetation cover management. Thus, the aim of this study was to (i) determine the effect of the cover management techniques (grazing, mowing and tillage) on the floristic composition in the cover vegetation in organic olive farms; and to (ii) evaluate the potential attraction between plant families composition in the SNH (cover and patch vegetation) and the Chrysopidae community. We hypothesized that vegetation cover management has a direct effect on plant communities and an indirect effect on the members of the Chrysopidae family that are attracted by particular plant families.

## 2. Results

### 2.1. Vegetation Indices

All vegetation indices show significant differences between the types of cover vegetation management (Figure 1 and Appendix A). Mowing had a significantly less total and cumulative plant cover than tillage and grazing (Figure 1a,b, and Appendix A) (GLMM total plant cover χ^2^ = 19.87, d.f. = 2, *p* < 0.001; cumulative plant cover χ^2^ = 12.66, d.f. = 2, *p* < 0.01). However, mowing was significantly richer and had a significantly higher diversity than grazing (Figure 1c,d, and Appendix A) (GLMM richness χ^2^ = 8.55, d.f. = 2, *p* < 0.05; LMM diversity χ^2^ = 6.05, d.f. = 2, *p* < 0.05).

Regarding the transect position in the cover vegetation, we found differences when it was between the rows and beneath the tree canopy (Figure 1 and Appendix A). The rows had higher total and cumulative plant cover than areas beneath the tree canopy (Figure 1a,b, and Appendix A) (GLMM total plant cover χ^2^ = 14.01, d.f. = 1, *p* < 0.001; cumulative plant cover χ^2^ = 10.24, d.f. = 1, *p* < 0.01). However, richness and diversity did not show significant differences for the transect position in the cover vegetation (Figure 1c,d, and Appendix A) (GLMM richness χ^2^ = 0.56, d.f. = 1, *p* = 0.45; LMM diversity χ^2^ = 0.15, d.f. = 1, *p* = 0.70).

In the patch vegetation indices, the percentage of total plant cover and cumulative plant cover were over 50% with no significant differences between farms (Appendix A) (GLMM total plant cover, χ^2^ = 6.71, d.f. = 4, *p* = 0.15; cumulative plant cover, χ^2^ = 5.33, d.f. = 4, *p* = 0.26). The patch vegetation in the La Pedriza was significantly richer than Los Almendros, Píñar (right) and Píñar (left) (GLMM richness, χ^2^ = 24.01, d.f. = 4, *p* < 0.001) (Appendix A). However, there were not significant differences in plant diversity (Appendix A) (LMM diversity, χ^2^ = 6.10, d.f. = 4, *p* = 0.19).

In the cover vegetation, we identified a total of 114 plant species from 18 families (Appendix A), where Malvaceae (13.07 ± 3.81), Fabaceae (12.27 ± 1.26) and Asteraceae (11.42 ± 0.91) had the largest percentage of plant cover (mean ± SE). Three members of the *Malvaceae* family were identified, of which the most relevant in terms of plant cover was *Malva nicaensis* All. (21.71 ± 5.96). Seventeen members of the family Fabaceae were identified, of which the species with the greatest plant cover were *Medicago minima* (L.) L. (21.71 ± 5.96) and *Trifolium tomentosum* L. (21.71 ± 5.96). Twenty-two members of the family Asteraceae were identified, of which the largest in terms of plant cover were *Carduus pycnocephalus* L. (30.00 ± 0.00), *Leontodon longirostris* (Finch & P. D. Sell) Talavera (23.40 ± 2.42) and *Crepis vesicaria* L. (15.37 ± 2.69). In the patch vegetation, 42 plant species from 20 families (Appendix A) were identified, dominated by the Rosaceae (61.17 ± 9.17), Fagaceae (34.96 ± 3.55)*,* Pinaceae (34.05 ± 5.49), Rhamnaceae (12.44 ± 4.87) and Poaceae (12.31 ± 1.72). The plant species with the greatest percentage of plant cover were *Prunus dulcis* (Mill.) D. A. Webb (Rosaceae) (61.17 ± 9.17), *Quercus rotundifolia* Lam. (Fagaceae) (41.34 ± 4.13) and *Pinus halepensis* Mill. (Pinaceae) (36.20 ± 5.54).

### 2.2. Interaction between Vegetation Indices, Vegetation Cover Management and Chrysopidae

The RDA showed that vegetation cover management and transect position had a significant effect on the plant community (Figure 2). Members of the family Fabaceae were mainly associated with grazing on both the beneath the tree canopy and row transects in 2016 (Figure 2a,b) (PERMANOVA row, Pseudo-F = 6.77, *p* < 0.001; PERMANOVA beneath the tree canopy, Pseudo-F = 10.72, *p* < 0.001). The families Asteraceae, Plantaginaceae and Brassicaceae were correlated with tillage, while the families Malvaceae, Convolvulaceae, Geraniaceae, Caryophyllaceae and Cistaceae were correlated with mowing (Figure 2a,b). Furthermore, Poaceae was associated with tillage in rows, and with mowing in beneath the tree canopy. In 2017, we also observed an effect of the vegetation cover management and transect position on the plant community following a similar pattern to the distribution of families in 2016 (Figure 2c) (PERMANOVA row, Pseudo-F = 10.31, *p* < 0.001; PERMANOVA beneath the tree canopy, Pseudo-F = 7.21, *p* < 0.001). Some families beneath the tree canopy had a higher correlation in 2017 (Figure 2c) than in 2016 (Figure 2a) such as Plantaginaceae in tillage, as well as Caryophyllaceae and Convolvulaceae in mowing. However, Brassicaceae was associated with tillage in 2016 (Figure 2a) and grazing in 2017 (Figure 2c).

Regarding the interaction between the botanical families and the family Chrysopidae, we observed that in the cover vegetation (Figure 3a) the chrysopid genus *Chrysoperla* was associated with the families Brassicaceae, Asteraceae, Rubiaceae, Apiaceae and Plantaginaceae, and that the genera *Apertochrysa* and *Pseudomallada* were mainly associated with the families Poaceae, Cistaceae, Convolvulaceae, Malvaceae, Caryophyllaceae and Geraniaceae (PERMANOVA, Pseudo-F = 7.98, *p* < 0.001). Furthermore, in the patch vegetation, the RDA (Figure 3b) revealed that the genus *Chrysoperla* was correlated with the families Rosaceae, Apiaceae, Asparagaceae and Caryophyllaceae, and that the genera *Apertochrysa* and *Pseudomallada* were associated with the families Oleaceae, Linaceae, Thymelaeaceae and Asteraceae (PERMANOVA, Pseudo-F = 5.79, *p* < 0.001).

## 3. Discussion

Our results showed that the type of vegetation cover management practiced produced significant differences in the composition and structure of the communities of plant species in the cover vegetation. The greater diversity values in the cover vegetation in the mowed farms could be due to the existence of a richer seed bank than in the other farms, as a consequence of the type of vegetation cover management (mowing) and its proximity to several extensive patches of repopulated vegetation [38]. The findings generated by previous studies of vegetation cover management techniques vary. For instance, our results are in agreement with Terzi et al. [45], who observed that both mowing and tillage are the most suitable vegetation cover management techniques for improving diversity in olive farms. However, Mast et al. [46] for fruit farms, and Simoes et al. [47] and Radić and Lakoš et al. [48] for olive farms, all found that only mowing increases diversity values in the cover vegetation. Regarding the transect position, we found that the rows had a higher total and cumulative plant cover, which may indicate that the mulching due to the pruning waste did not have a significant effect. Furthermore, there were not differences between the transect position for the richness and diversity.

In terms of plant cover by species, we observed that *C. vesicaria* had greater cover values in tilled and mowed olive farms than in grazed olive farms, while *L. longirostris* had greater cover values in grazed olive farms. This is due to the ability of some plant species to withstand trampling and continuous defoliation by animals [49], which leads to the selection of plant species with tough, long-living leaves, that are adapted to regrowth. *Crepis vesicaria* is a scapose hemicryptophytic plant species with erect stems up to 100 cm, with leaves distributed along its scape. *Leontodon longirostris* is a rosulate therophyte, smaller in size and with leaves pressed to the ground that allow for less leaf consumption. Furthermore, *C. vesicaria* was also more present in the tilled olive farms, probably because the tillage was not deep enough to damage its roots.

Our results showed greater plant cover values in zoochoric plant species in grazed olive farms. Previous studies of zoochory have concluded that perennial plant species, with large leaf areas and low leaf dry-matter content, are selected over small plant species with light seeds and early flowering periods [50]. Good examples of this case from our study are *M. minima* and *M. rigidula*, both creeping therophytes with legume sizes of 2.5–3.5(4) mm in the former and 4.5–7 × 6.5–7.5 mm in the latter [51]. Although these zoochoric species are also present in mowed olive farms, their plant cover values are lower. A special case is *M. polymorpha*, which is also a zoochoric plant species but with larger fruit (1.5–9.5 × 4–9 mm) than either *M. minima* or *M. rigidula* [51]. We recorded greater plant cover values in tilled than in grazed olive farms for *M. polymorpha*. This is possibly due to the fact that this species is intolerant of trampling by livestock, a variable dynamic that is dependent on meteorological conditions [52]; as well, plant height is negatively correlated with grazing [53].

The redundancy analysis determined how vegetation cover management affects plant species community composition in the cover vegetation. In both studied zones, the rows and beneath the tree canopy, the Fabaceae family was seen to be favoured by grazing. The pastoral paradox is that well-managed grazing promotes the abundance of the most consumed species such as leguminous plants [54], especially when grazing is carried out by sheep [55]. The families Asteraceae and Plantaginaceae, on the other hand, are associated with tillage, probably because their seed banks are mobilised by tilling, which brings seeds to the surface and allows them to germinate [56]. We should also highlight the greater plant cover values for *Anacyclus clavatus* (Desf.) Pers. in the tilled olive farm. Previous studies have shown that *A. clavatus* in crops is favoured by no or minimal tillage [57,58]. Furthermore, the genus *Anacyclus* has heterocarpia with a high capacity for dispersal and germination at different times of the year, with different types of achenes (winged and non-winged) [59,60]. Thus, *A. clavatus* could have increased the plant cover in the tilled olive farm, where the tillage continuously mobilises the seed bank over the years.

Finally, mowing allowed the families Geraniaceae, Malvaceae and Convolvulaceae to dominate in our study area, probably because their low size and creeping habits are not damaged by mowing and thus they develop better than more upright species [61]. Furthermore, a previous study shows that broad-leaved plant species covered by crop residues under mowed management do not stop growing and even increase their plant cover [62]. This might explain the greater values in mowed olive farms of *Erodium malacoides* (L.) and *Convolvulus arvensis* L., both of which are broad-leaved and have creeping habits with resprouting capacities.

In grazed farms, both diversity and richness were lower than in mowed farms. Nonetheless, Moonen and Bàrbieri [63] stated that a general increase of biodiversity will not have a positive impact on the specific agroecosystem processes, their environmental impact or sustainability. On the contrary, they recommended to increase functional biodiversity, i.e., to increase diversity within functional groups to promote agroecosystem processes. However, grazed farms did not have lower levels of total plant cover. Continuous grazing for several years by sheep has led to the selection of the most palatable and adapted families such as the Fabaceae, Asteraceae and Poaceae, and has stabilized the floristic composition and diversity values in this agroecosystem [55]. Similar results have been obtained in almond farms by Ramos-Font et al. [64], who found that diversity and richness were greater under managed tillage, even though most species could be considered to be ‘weeds’; by comparison, under grazing, although there were fewer species, they belonged to communities that were characteristic of therophytic subnitrofilous pastures.

Plant-insect interactions are linked to feeding—i.e., plants as floral resources (pollen and honeydew) and arthropods as prey for predators or hosts to parasitoids—and to shelter and breeding sites [65,66,67]. The degree of preference of Chrysopidae for certain plant substrates is one of the most notable aspects of their biology [23]. Our results give a broad picture of the relationship between Chrysopidae genera and botanical families in olive farms, and help to determine which botanical families favour the presence of chrysopids and could be most suitable for implementing biological control through conservation. Previous studies have shown that Chrysopidae are attracted by plant species from the families Asteraceae, Brassicaceae, Fabaceae and Malvaceae, which they use in olive farms as reproduction sites [12]. Furthermore, Villa et al. [20] found that Chrysopidae feed on herbaceous plant species belonging to the families Asteraceae, Apiaceae and Brassicaceae, which are well represented in our olive farms. It is known that herbaceous species from the four former families provide food resources such as honeydew and pollen to palyno-glycophagous Chrysopidae species that allow them to survive more than 20 days and produce eggs [24,68,69]. Although, the nectaries of the glossy flowers of the Asteraceae are apparently not as readily available as those of the Apiaceae, Brassicaceae and Geraniaceae, Chrysopidae adults consume pollen on Asteraceae plant species [24,70]. Furthermore, Chrysopidae adults feed on pollen deposited on the vegetation surface [20,24,71]. The family Plantaginaceae appears in the RDA analysis to be related to the presence of the genus *Chrysoperla*. The *Plantago* spp. that we identified in the cover vegetation do not have nectaries but are early-flowering and anemogamous, which may affect their exploitation by Chrysopidae. Villa et al. [20] found that the pollen of these species was also consumed by these insects. Interestingly, pollen from *Plantago lanceolata* L., a plant that is favoured by tillage and is not found in grazed or mowed olive farms, was found by Nunes Morgado et al. [70] to be the most frequent pollen in Chrysopidae guts.

The results for the diversity, abundance and distribution of botanical families in the patch vegetation concur with the potential vegetation of the study area, where the dominant vegetation series in the landscape farms is meso-Mediterranean holm oak *Paeonio-Querco rotundifoliae sigmetum* [43,44]. Most of these vegetation patches correspond to degraded holm oak forests in which *Q. rotundifolia* and *Quercus coccifera* L. are the predominant native plant species. *Cistus albidus* L. and *C. clusii* Dunal are the serial stages of these holm oak groves, accompanied by *Thymus zygis* L. subsp. *gracilis* (Boiss.) R. Morales in semi-natural habitats on lithosols. Likewise, in the drier areas there are communities of grasses dominated by *Macrochloa tenacissima* (L.) Kunth) and *Brachypodium retusum* (Pers.) P. Beauv. subsp. *retusum*, in which there are some *Labiatae* spp. such as *Rosmarinus officinalis* L. The presence of *P. dulcis* and *P. halepensis* is due to the crops associated with olive farms and previous reforestation events, respectively, carried out along the boundaries and in the patches of vegetation adjacent to the olive farms. It is known that some Chrysopidae species are closely associated with *Q. rotundifolia* and inhabit Mediterranean Iberian holm oak forests [23]. Previous studies have shown that woody species such as *P. halepensis*, *P. dulcis* and *Q. rotundifolia* in the patch vegetation are used as reproduction and refuge sites [11,15]. Furthermore, Villa et al. [20] observed that Chrysopidae feed on anemophilous and entomophilous trees and scrub families such as Oleaceae, Cistaceae, Pinaceae, Fabaceae and Ericaceae. It is worth underlining the possible importance of the flowering phenology of some of the species belonging to this group of botanical families. Pollen and nectar become available when *P. dulcis* flowers, normally in February-March in these latitudes, followed by *C. albidus* and *C. clusii*, which start flowering in March-April. During the months of April-May-June, the remaining species in the cover vegetation come into flower. This would imply that in some of these plant species forming part of the vegetation patches temporal staggering and overlapping in the food supply for Chrysopidae occurs.

## 4. Materials and Methods

### 4.1. Study Area

The study was conducted between May 2016 and October 2017 in five organic olive farms (*Olea europaea* L. var Picual) in the Montes Orientales (Granada Province, Spain; Table 1). The main land use in this area is olive cultivation, which covers 49,927 hectares (ha) [72]. All farms are located at a similar altitude (800–1000 m a.s.l.) and have surface areas between 0.9 and 215 ha and a plantation framework of either 8 × 8 m or 12 × 12 m; the farms are between 0.8 and 13 km apart (Table 1). The growing season in the Spanish olive farms start in February and finishes the next year after the harvest period, which is carried out between October and February [27]. All our farms had spontaneous vegetation cover, which had been removed by grazing, mowing or tillage at one time between April–May (Table 1). In addition, during the post-harvest period (February–September), crushed pruning waste was placed along the centre of the rows as mulch. The patch vegetation is a mixed natural remnant from the meso-Mediterranean *Paeonio-Querco rotundifoliae sigmetum* vegetation series, sown by the farmers and/or reforestation events. *Quercus rotundifolia* is the dominant native plant species in the vegetation series. *Crataegus monogyna* Jacq., *Rhamnus lycioides* L. or *Q. coccifera* are the frequent serial and sub-serial stages in the vegetation series, accompanied by rosemary (*R. officinalis*), asparagus (*M. tenacissima*) and thymes (*T. zygis*). The spatial distribution across the farms is highly variable between patch and linear which acts as hedgerow. We did not record fire events in the patch vegetation in the farms selected that could affect to the flora and vegetation sampling during the sampling period.

The damage caused by pests on olive trees can be divided into immediate damage with repercussions on the harvest of the current crop year such as attacks on buds, flowers and fruits, or delayed attacks, those which carry over into other seasons such as attacks on shoots, branches, trunk and roots. The losses by pests and diseases comprise around 30% of olive yield. Of this, about one-third are caused by two pests, *Bactrocera oleae* (Rossi, 1790) and *Prays oleae* (Bernard, 1788) [73]. Although there was no incidence of the pest *B. oleae* in the farms studied, the incidence of the pest *P. oleae* and the disease *Fusicladium oleagineum* (Castagne) Ritschel & U. Braun was remedied by timely and targeted treatments (one for the pest in local patches and two for the disease) with *Bacillus thuringiensis* var. *kurstaki* (strain PB54) and copper oxychloride (50–70% *w*/*v*), which are listed in Annex II of Commission Regulation (EC) no. 889/2008 concerning organic management [74]. Silva et al. [75] classified the copper oxychloride in a bioassay as harmless to Chrysopidae adults and pupae according to the International Organization for Biological and Integrated Control of Noxious Animals and Plants (IOBC) classification. Soils are calcic cambisol, with calcaric regosol, leptosol and rendsinas [76]. The weather is characterized by an oceanic-pluviseasonal Mediterranean bioclimate, with an upper meso-Mediterranean thermotype and a low subhumid ombrotype, an annual mean temperature of 15.1 °C and cumulative mean precipitation of 592 mm [77,78] (Appendix A).

### 4.2. Flora and Vegetation Sampling

We used the non-destructive point quadrat method [79] during the sampling of the cover vegetation. This method records the presence or absence of herbaceous species determined by contact with a 2-mm-diameter needle at 100 points, five centimetres apart, along a 5 m transect (Figure 4). Eight transects per olive farm were established in May 2016 and 2017 before the management of the cover vegetation: four transects one meter away from the tree trunks (i.e., beneath the olive tree canopy) and four transects in the middle of the rows of trees, with a minimum distance of 25 m between transects. The transect position (in the middle of the rows and beneath the olive tree canopy) was performed to evaluate the mulching effect due to the crushed pruning wastes in the row. In 2017 for Los Almendros farm, we have only four transects since the farmer tilled the soil before sampling in the middle of the rows.

In the patch vegetation, which consists mainly of trees and shrubs and is not affected by vegetation cover management, we could sample in any time of the year and we used the line intercept method as the most suitable methodology for woody communities [80] to record the interception range in centimetres of plant species along a 25 m transect. A total of 60 transects, three transects per patch vegetation area, with a 50 m gap between consecutive transects and a minimum distance of 150 m between each patch vegetation, were carried out in October 2017 (Figure 4). The number of patches examined per olive farm varied from two to six depending on the farm surface area (Norberto two patches; La Pedriza three patches; Píñar (left) four patches; Píñar (right) five patches and Los Almendros six patches).

The vegetation indices obtained by these two methodologies were calculated as follows:Total plant cover as the percentage of soil covered by vegetation.Cumulative plant cover as the sum of the cover of each plant species, expressed as a percentage.Plant cover by species as the percentage of soil covered by the plant species *i*. The cover vegetation (*S.Cov_cover_*) (Equation (1)) is expressed as the number of contacts per species *i* divided by the total contacts along the transect (100). The patch vegetation (*S.Cov_patch_*) (Equation (2)) is expressed as the intercept range in centimetres per species *i* divided by the total transect length in centimetres (2500 cm).
(1)$$S.\;Cov_{cover}=\frac{number\;of\;contact\;per\;species_{i}}{total\;transect\;contact}\times 100$$
(2)$$S.\;Cov_{patch}=\frac{interception\;range\;by\;species_{i}}{total\;transect\;length}\times 100$$Plant cover by family as the sum of the percentage of the soil covered by the plant species belonging to family *i*.Richness as the number of plant species per transect.Diversity was calculated using the Shannon index ($H_{cover\;or\;patch}^{\prime }$) [81] (Equation (3)), where *P_i_* is the relative frequency per species *i* and S is the number of species recorded. In the cover vegetation, *P_i_* is the number of contacts per species *i* divided by the total number of contacts for all species recorded (Equation (4)). In the patch vegetation, *P_i_* is the interception range in centimetres per species *i* divided by the total interception range in centimetres for all species recorded (Equation (5)).
(3)$$H_{cover\;or\;patch}^{\prime }=-\overset{S}{\underset{i=1}{\sum }}P_{i}\times lnP_{i}$$
(4)$$P_{i\;cover}=\frac{number\;of\;contacts\;per\;species_{i}}{\sum number\;of\;contacts\;for\;all\;species}$$
(5)$$P_{i\;patch}=\frac{interception\;range\;per\;species_{i}}{\sum interception\;range\;for\;all\;species}$$

Plants that were difficult to identify in situ were labelled and identified in the laboratory using the keys in the *Flora Vascular de Andalucía Oriental* [82].

### 4.3. Chrysopidae Sampling

The arthropod community of olive farms located in south Spain is highly active from April to October [31,83]. Chrysopidae collection was undertaken fortnightly between April and October 2016, with a total of 13 sampling events, in the same olive farms that were used to obtain the flora and vegetation samples. A total of 75 trees per species, almond, oak and pine, per sampling in the patch vegetation, and 75 olive trees within the farms were randomly selected from five olive farms depending on their availability, with a minimum distance of 15 m between trees (Figure 4). We tried to collect Chrysopidae individuals in the cover vegetation in some of the farms studied; however, we collected few individuals, and we decided not to sample the cover vegetation in all farms studied. Tree canopies were vacuumed using a field aspirator (InsectaZooka, BioQuip^®^, Rancho Dominguez, CA, USA) for two minutes; suction samples were cold-stored in the field and frozen at −20 °C until identification. Chrysopidae adults were counted and identified up to species level following Monserrat [84], Duelli et al. [85], Canard et al. [86] and Breitkreuz et al. [87]. Chrysopidae data were grouped at genus level to get a broader idea about their relationship with plant family community in the SNHs. A detailed description of the design and sampling for Chrysopidae can be found in Alcalá Herrera and Ruano [18].

### 4.4. Statistical Analyses

We analysed the data using R version 3.6.3 [88] and R Studio version 1.1.456 [89], together with the packages *glmmTMB* [90], *lme4* [91], *Matrix* [92], *vegan* [93] and *climatol* [94]. Residuals were examined for model validation using the *DHARMa* package [95]; we also checked fixed factors for significance using Wald tests with the *car* package [96] and multiple comparison between fixed factors for significance using *lsmeans* [97].

To investigate the effect of the vegetation cover management techniques on the floristic composition in the cover vegetation, the vegetation indices (total plant cover, cumulative plant cover, richness and diversity) obtained by point quadrat methodology were analysed using generalized linear mixed models (GLMMs) and a linear mixed effect model (LMM). In each model, the vegetation cover management was established as a fixed factor, and the sampling date, olive farm and transect were established as random factors to account the variations on time and spatial location. We have used a GLMM with a beta-binomial distribution for the total plant cover and cumulative plant cover, a GLMM with a Poisson distribution for the richness and a LMM with a Gaussian distribution for the diversity.

To analyse the landscape homogeneity across the patch vegetation sampled, the vegetation indices (total plant cover, cumulative plant cover, richness and diversity) were analysed using GLMMs with beta-binomial and Poisson distribution, as well as a LMM with Gaussian distribution. In each of these models, the olive farm was included as fixed factor and the patch vegetation in each olive farm was set as a random factor to correct local spatial variations and the number of patches of vegetation sampled.

A redundancy analysis (RDA), with a permutational multivariate analysis of variance (PERMANOVA) using a Bray–Curtis distance and 999 permutations, was performed to test whether plant cover by family in the rows and beneath the olive tree canopy differed between vegetation cover management types (grazing, mowing and tillage). In Los Almendros farm, we omitted the analysis of the rows due to the farmer having tilled the soil before sampling in the middle of the rows. Furthermore, we carried out a RDA to establish how the most representative Chrysopidae genera collected were related to the plant cover at family level in the SNHs (cover vegetation and patch vegetation).

## 5. Conclusions

We observed that the vegetation cover management techniques have a direct effect on plant families and species composition, where the most representative plant families were Asteraceae (*A. clavatus*, *C. vesicaria* and *L. longirostris*) and Fabaceae (*M. minima* and *M. rigidula*). While the landscape vegetation was similar in all the farms studied and the *Paeonio-Querco rotundifoliae sigmetum* was the dominant vegetation series with *Q. rotundifolia* (Fagaceae) as the most notable species, together with plant species from reforestation by *P. halepensis* (Pinaceae) and other species such as *P. dulcis* (Rosaceae) dating from previous agronomic practices. Furthermore, the mulching did not have an effect in the plant composition as there were not significant differences between the transect positions in richness and diversity. In terms of the interaction between the Chrysopidae and the plant communities, the genus *Chrysoperla* was related to the families Plantaginaceae, Brassicaceae and Asteraceae in the cover vegetation, as well as to the Apiaceae, Caryophyllaceae, Asparagaceae and Rosaceae in the patch vegetation in semi-natural habitats. The genera *Apertochrysa* and *Pseudomallada* were associated with the families Malvaceae, Poaceae and Geraniaceae in the cover vegetation, as well as to the Cupressaceae, Poaceae, Pinaceae and Cistaceae in the patch vegetation in semi-natural habitats. Our study has important implications for farmers concerned with biodiversity conservation and agricultural landscape as it shows that the vegetation cover management techniques have a potential to select plant families in the cover vegetation. Furthermore, farmers may increase the presence of plant species with seeding or planting with autochthonous plant species in the cover or patch vegetation to increase the biodiversity and richness in the agroecosystem. Further research linking potential effects of specific plant species or type of cover vegetation on the arthropod community would clearly help in understanding their specific roles in the life cycle of the arthropod community in support of pest control in the olive agroecosystem.

## Figures and Tables

**Figure 1 plants-11-03255-f001:**
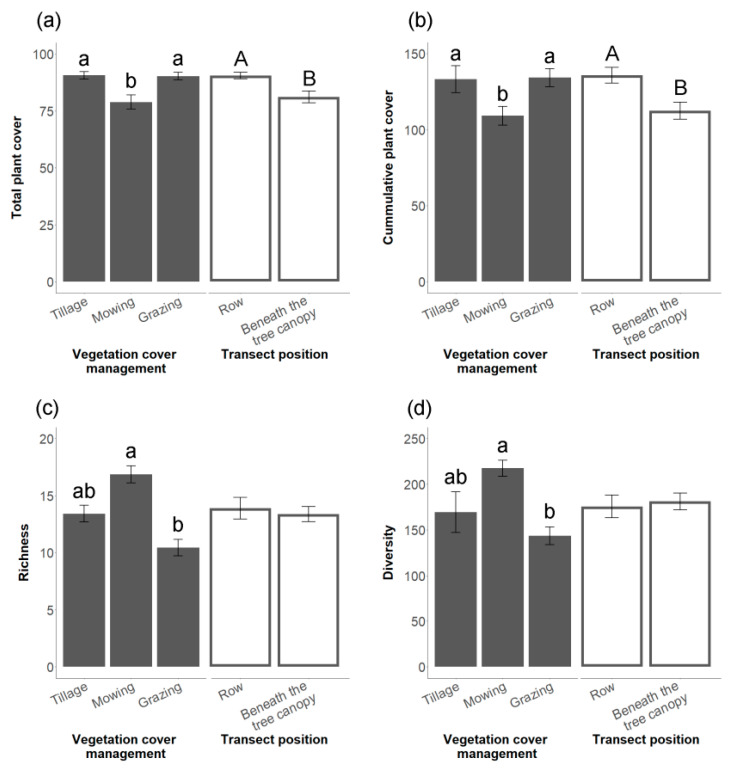
Vegetation indices (mean ± SE): total plant cover (%), cumulative plant cover (%), richness and diversity of the cover vegetation (**a**–**d**), respectively, for the vegetation cover management and transect position. Different letters indicate statistically significant differences between the vegetation cover management or transect position (*p* < 0.05).

**Figure 2 plants-11-03255-f002:**
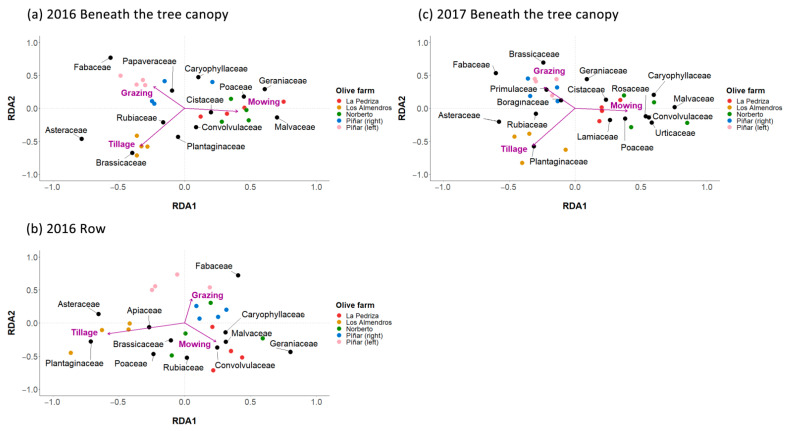
Redundancy analysis (RDA) biplot representing plant cover per family (%) in each transect in the cover vegetation between the rows and beneath the tree canopy subject to different vegetation cover management in 2016 (**a**,**b**), respectively, and beneath the tree canopy in 2017 (**c**). Vegetation cover management is represented by purple arrows, olive farms by coloured dots and plant families by black dots.

**Figure 3 plants-11-03255-f003:**
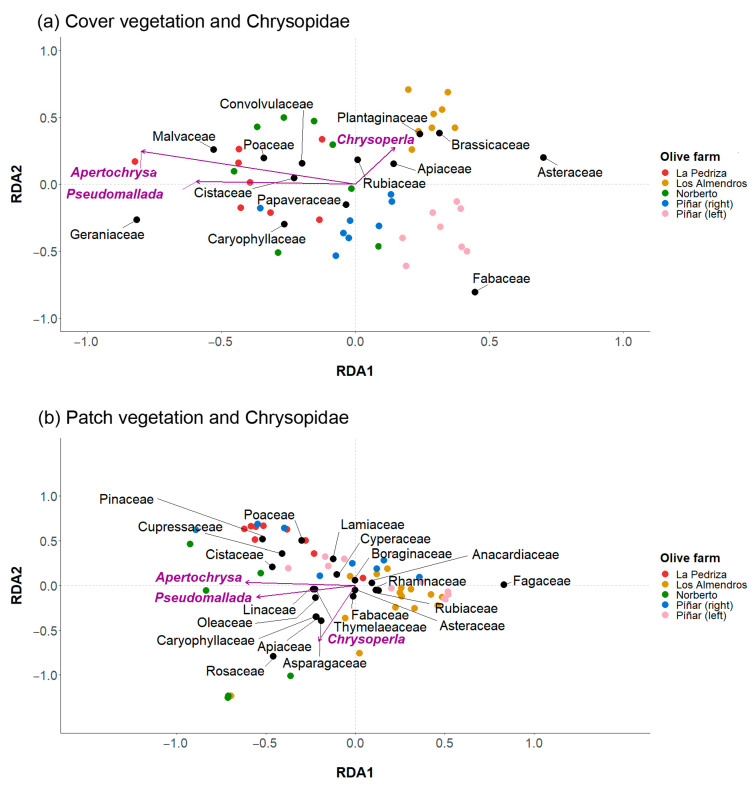
Redundancy analysis (RDA) biplot representing plant cover per family (%) in each transect and the number of Chrysopidae genera collected in each olive farm (**a**) in the cover vegetation and (**b**) in the patch vegetation. Chrysopidae genera are represented by purple arrows, olive farms by coloured dots and plant families by black dots.

**Figure 4 plants-11-03255-f004:**
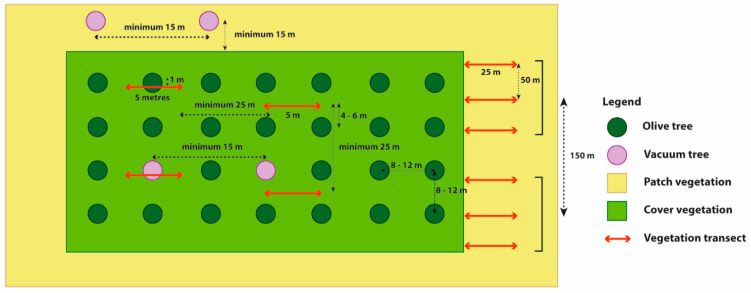
Experimental design diagrams in each farm for flora and vegetation, and Chrysopidae sampling.

**Table 1 plants-11-03255-t001:** Characteristics of the studied olive farms: geographical parameters, altitude, surface area, plantation framework and vegetation cover management.

Olive Farm	Coordinates (datum: WGS 84)	Altitude (m)	Area (ha)	Plantation Framework (m)	Vegetation Cover Management
La Pedriza	37°20′17.44″ N; 3°33′39.21″ W	954	0.9	8 × 8	mowing
Los Almendros	37°22′24.76″ N; 3°37′46.03″ W	904	215	8 × 8	tillage
Norberto	37°19′5.96″ N; 3°34′9.92″ W	1009	4.3	8 × 8	mowing
Píñar (right)	37°24′14.29″ N; 3°29′14.13″ W	899	58	12 × 12	grazing
Píñar (left)	37°24′40.93″ N; 3°28′52.41″ W	895	124	12 × 12	grazing

## Data Availability

The authors declare that all data supporting this study’s findings are provided in the article and in the Appendix A.

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
