# Peer review of "Vegetation Cover Management and Landscape Plant Species Composition Influence the Chrysopidae Community in the Olive Agroecosystem"

_plants, 2022, doi:10.3390/plants11233255_

Round 1

Reviewer 1 Report

This is an interesting paper with new data on the biological management of olive orchards. I suggest only a few changes.

Author Response

Reviewer 1. Comments and Suggestions for Authors

This is an interesting paper with new data on the biological management of olive orchards. I suggest only a few changes.

Response: Thank you for all your comments and for taking to time to review the manuscript. We have taken into consideration all your comments and have made changes in the abstract, keywords, introduction, and result sections.

Line 26. explain better (ex.: collected by a suction trap...)

Response: We have modified the text accordingly (line 26).

Line 41. add: Pseudomallada

Response: We have modified the text accordingly (line 42).

Line 68. some further information on the importance of Chrysopidae as predators; are they generalists or specialists?

Response: We have modified the text accordingly (lines 71 to 72).

Line 145. please, check it

Response: We have checked the information (line 148).

Line 145. longirostris

Response: We have modified the text accordingly (line 148).

Line 207. longirostris

Response: We have modified the text accordingly (line 212).

Line 212. longirostris

Response: We have modified the text accordingly (line 217).

Line 348-349. has it influence on the Chrysopid life cycle?

Response: Silva et al. 2006 evaluate the action of copper oxychloride on pupae and adults of Chrysoperla externa. The bioassays classified the copper oxychloride as harmless to adults and pupae according the IOBC classification. We have modified the text (lines 356 to 359).

Silva, R. A.; Carvalho, G, A.; Carvalho, C. F.; Reis, P. R.; Souza, B.; Pereira, A. M. A. R. (2006). Action of pesticides used in coffee crops on the pupae and adults of Chrysoperla externa (Hagen, 1861) (Neuroptera: Chrysopidae). Ciência Rural, Santa Maria, 36: 1, 8-14. DOI: 10.1590/S0103-84782006000100002

Line 448. I suggest to eliminate this paragraph because it does not give information not already presented in the Discussion

Response: Thank you for your comment but we prefer to keep the conclusion section. In our view, it summarizes the knowledge acquired through the interpretation and discussion of the results in our study.

Reviewer 2 Report

I just finished to review the manuscript Plants-2003607 with the title: Vegetation cover management and landscaper plant species composition influence the Chrysopidae community in olive agroecosystem. 

I food the manuscript very well organized and with a robust statistical approach. My only comments are referred to some mistakes in the bibliography. I was also surprised that the name of the Families (for the plants) are written in italics, differently from what is the normal situation in almost all the scientific papers. In the text, the insect families are not reported in italics... so I assuming that it is not the Plants journal rule. 

Returning back to the manuscript context, for my personal experience I found that overgrazing can induce dramatic differences in the composition (qualitative and quantitative) of the cover: some of the most unsuitable (because poisonous or unpalatable) species become much more common and are making dramatic changes in the habitat, against the suitable plant species. I am wondering if authors recorded similar situations in their field plots. 

Author Response

Reviewer 2. Comments and Suggestions for Authors

I just finished to review the manuscript Plants-2003607 with the title: Vegetation cover management and landscaper plant species composition influence the Chrysopidae community in olive agroecosystem.

I food the manuscript very well organized and with a robust statistical approach. My only comments are referred to some mistakes in the bibliography. I was also surprised that the name of the Families (for the plants) are written in italics, differently from what is the normal situation in almost all the scientific papers. In the text, the insect families are not reported in italics... so I assuming that it is not the Plants journal rule.

Response: Thank you for all your comments and for taking to time to review the manuscript. We have modified the Families format to the plants in plain through the text. Insect families must be reported in plain format.

Returning back to the manuscript context, for my personal experience I found that overgrazing can induce dramatic differences in the composition (qualitative and quantitative) of the cover: some of the most unsuitable (because poisonous or unpalatable) species become much more common and are making dramatic changes in the habitat, against the suitable plant species. I am wondering if authors recorded similar situations in their field plots.

Response: Thank you for your comments. The grazing was managed by a shepherd around the whole farm in a specific period of time. The shepherd tries to avoid a overgrazing on the cover vegetation. Regarding to the poisonous or unpalatable plant species in the cover vegetation, you could find a complete list of plant species identified in the supplementary material. In our field plots we did not recorded a similar situation as you.

Line 346. The fungi disease Fusicladium oleagineum (Castagne) Ritschel & U. Braun.

Response: We have modified the text accordingly (line 353).

Line 503. I am not sure it should be in italics.

Response: We have modified the text accordingly (line 513).

Line 505. Text in bold.

Response: We have modified the text accordingly (line 515).

Line 535. Text in italics.

Response: We have modified the text accordingly (line 545).

Line 565. Text in bold.

Response: We have modified the text accordingly (line 580).

Line 572. Text in bold.

Response: We have modified the text accordingly (line 587).

Line 592. Text in bold.

Response: We have modified the text accordingly (line 607).

Line 598. Text in bold.

Response: We have modified the text accordingly (lines 613 to 614).

Lines 622. Text in bold and is there anything missing (i.e. number of pages)?.

Response: We have modified the text accordingly (lines 637 to 638).

Line 632. In italics and the species name not uppercase.

Response: We have modified the text accordingly (line 648).

Line 636. Text in bold.

Response: We have modified the text accordingly (line 652).

Line 641. Text in bold.

Response: We have modified the text accordingly (line 657).

Line 661. Text in bold.

Response: We have modified the text accordingly (line 677).

Line 664. Text in bold.

Response: We have modified the text accordingly (lines 680 to 681).

Line 669. Text in bold.

Response: We have modified the text accordingly (line 685).

Lines 670 to 671. Italics and the species name not in uppercase.

Response: We have modified the text accordingly (lines 686 to 687).

Line 675. Text in bold.

Response: We have modified the text accordingly (line 691).

Line 678. Text in bold.

Response: We have modified the text accordingly (line 694).

Line 681. Text in bold.

Response: We have modified the text accordingly (line 697).

Line 692. Bold and the number of pages is missing.

Response: We have modified the text accordingly (line 712).

Line 698. Text in bold.

Response: We have modified the text accordingly (line 719).

Lines 704 to 709. Bold and the species name should not be in uppercase

Response: We have modified the text accordingly (lines 725 to 727).

Line 712. Text in bold.

Response: We have modified the text accordingly (line 733).

Line 718. Text in bold.

Response: We have modified the text accordingly (line 739).

Line 721. Text in bold.

Response: We have modified the text accordingly (line 742).

Line 723. Text in bold.

Response: We have modified the text accordingly (line 744).

Line 724. Text in bold.

Response: We have modified the text accordingly (line 745).

Reviewer 3 Report

In the manuscript entitled “Vegetation cover management and landscape plant species composition influence the Chrysopidae community in olive agroecosystem” authors try to evaluate the effect of vegetation management (mowing, tillage, grazing) on floristic composition and on Chrysopidae.

The topic of the research is interesting even if the decision (not sufficiently justified in the manuscript) of the authors to focus only on Chrysopidae strongly limits the possible positive effects of the research for the management of the olive grove. In fact, the role of parasitoids in the biological control of key olive pests (Bactrocera olae and Prays oleae) seems clearly more important than that offered by Chrysopidae. Nevertheless the authors did not consider the presence of parasitoids (or other predators) although the used sampling method (vacuum) was adequate for the collection of these insects.

Anyway also the object to evaluate the effect of cover management on Chrysopidae is not possible with this experimental design. The number of olive farms is too low: only five farms to study three different types of cover management. Tillage is present in only one farm. The great variability in farm characteristics (Table 1 and Results in Herrera and Ruano, 2022) does not allow to really disentangle cover management from other possible effect. For example:

1)     The Plantation framework in grazing olive farms (Pinar) is different from mowing and tillage. So differences is due to grazing or different plantation framework (more space between plants)?

2)     The dimension of farms with mowing is clearly lower than other farms: 0.9-4.3 instead of 58-215 ha. This means that tillage and grazing olive fields are 13-239 greater than mowing fields. Therefore differences are due to small dimension (with possible greater effect of landscape) or to mowing?

3)     Two farms (Los Almendros and and Pinar left) has no pine in the surrounding (at least following the results in Herrera and Ruano, 2022). Since Pine support a lower population of Chrysopidae (again following the interesting results in Herrera and Ruano, 2022) this can have a great effect on Chrysopidae population.

4)     Tillage management has only one olive farm and (lines 367-368) in 2017 the sampling in this field has problem (only four transect instead of eight as in other fields). So we have only one field with critical data for one of the two sampling year: it is not correct to add this management in 2017 analysis.

In addition present research try to study the indirect effect of cover management on Chrysopidae though the effect of floristic composition in the olive grove. However no data are given of the intermediate state in this sequence; in particular no information is given about preys (both target or additional), other predators (with possible intraguild predaction effect), even if vacuum sampling allow to obtain these data. These information are important to understand if the supposed attraction of some plant is a per se effect or is due to change in other insect population.

I’m not a mother language, anyway I think that English check of the manuscript is necessary.

Author Response

Reviewer 3. Comments and Suggestions for Authors

In the manuscript entitled “Vegetation cover management and landscape plant species composition influence the Chrysopidae community in olive agroecosystem” authors try to evaluate the effect of vegetation management (mowing, tillage, grazing) on floristic composition and on Chrysopidae.

The topic of the research is interesting even if the decision (not sufficiently justified in the manuscript) of the authors to focus only on Chrysopidae strongly limits the possible positive effects of the research for the management of the olive grove. In fact, the role of parasitoids in the biological control of key olive pests (Bactrocera olae and Prays oleae) seems clearly more important than that offered by Chrysopidae. Nevertheless the authors did not consider the presence of parasitoids (or other predators) although the used sampling method (vacuum) was adequate for the collection of these insects.

Response: Thank you for all your comments and for taking to time to review the manuscript. Chrysopidae is one of the most studied predators in olive orchard which is present in olive orchards all year round, plays a key role in the predations of the eggs and larvae of the olive moth, Prays oleae (Bernard, 1788), one of the principal olive pests (Campos, 1989; Ramos et al., 1984a; Szentkirályi, 2001). Chrysoperla carnea sensu lato (Stephens, 1836) was declared insect of the year in 1999 (Dathe, 1999) due to its agricultural importance. The adult stage of most species, which depends on the vegetation to feed on nectar, pollen and honeydew, has a palyno-glycophagous diet (Devetak and Klokocovnik, 2016; Villenave et al., 2006). It is important to evaluate the potential attraction of the floristic composition in the cover vegetation and patch vegetation. Furthermore, there was not incidence to Bactrocera oleae (Rossi, 1790) in the farms studied (lines 349 to 350). Although, we appreciate your comments to consider parasitoids and other predators in the present study, in our view we did not include on it. We have justified focus only on Chrysopidae in the introduction section (line 44 to 111). We have modified the text (lines 44 to 111).

Campos, M., 1989. Observaciones sobre la bioecologia de Chrysoperla carnea (Stephens) (Neuroptera: Chrysopidae) en el sur de España. Neuroptera Int. 5, 159–164.

Ramos, P., Campos, M., Ramos, J.M., 1984a. Estabilización del ataque de Prays oleae Bern. y de la actividad de los depredadores oófagos sobre el fruto del olivo. Boletín Sanid. Veg. Plagas 10, 239–243.

Szentkirályi, F., 2001a. Lacewings in vegetables, forests, and other crops, in: Whittington, A.E., McEwen, P.K., New, T.R. (Eds.), Lacewings in the Crop Environment. Cambridge University Press, Cambridge, UK, pp. 239–292. https:// doi.org/DOI: 10.1017/CBO9780511666117.012.

Dathe, H.H., 1999. Das erste Insekt des Jahres: die Florfliege Chrysoperla carnea (Stephens, 1836) (Neur., Chrysopidae). Entomol. Nachr. Ber. 43, 1–3.

Devetak, D., Klokocovnik, V., 2016. The feeding biology of adult lacewings (Neuroptera): a review. Trends Entomol. 12, 29–42.

Villenave, J., Deutsch, B., Lode, T., Rat-Morris, E., 2006. Pollen preference of the Chrysoperla species (Neuroptera: Chrysopidae) occurring in the crop environment in western France. Eur. J. Entomol. 103, 771–777.

Anyway also the object to evaluate the effect of cover management on Chrysopidae is not possible with this experimental design. The number of olive farms is too low: only five farms to study three different types of cover management. Tillage is present in only one farm. The great variability in farm characteristics (Table 1 and Results in Herrera and Ruano, 2022) does not allow to really disentangle cover management from other possible effect. For example:

1) The Plantation framework in grazing olive farms (Pinar) is different from mowing and tillage. So differences is due to grazing or different plantation framework (more space between plants)?

Response: Although the plantation framework between olive trees in grazed farms (12 x 12 m) is larger than mowed (8 x 8 m) and tilled (8 x 8 m), the flora and vegetation sampling methodology was the same in all farms. Transects were taken one meter away the tree trunks and in the middle of the rows (lines 368 to 378). In all row farms there is enough space between olive trees to the emergence of the cover vegetation. We were worried about the mulching effect of the crushed pruning waste placed along the middle of the rows than the four-meter difference in the plantation framework. However, our results showed that total plant cover and cumulative plant cover were higher in the row than beneath the tree canopy. Therefore, plantation framework was not an issue.

2) The dimension of farms with mowing is clearly lower than other farms: 0.9-4.3 instead of 58-215 ha. This means that tillage and grazing olive fields are 13-239 greater than mowing fields. Therefore differences are due to small dimension (with possible greater effect of landscape) or to mowing?

Response: The farm selection was a combination of pesticide management (organic), vegetation cover management (mowing, tillage and grazing), the available of the tree species in the patch vegetation (almond, oak and pine) and weather conditions. Although Andalusia is a world leader in terms of the surface area devoted to olive farms (lines 75 to 77), it was difficult to find farms with the specific combination of features. Furthermore, the 53% of the olive farms in our study area has a surface lower than 5 ha.

We are agreeing with you that there is difference in the surface area in the farm studied. However, we evaluate the flora and vegetation (lines 364 to 409) with the same methodology in all of them, as well as Chrysopidae was sampled in almond, oak and pine trees, which were located in the patch vegetation, and olive trees within the orchard (lines 410 to 427). Furthermore, we have observed that the patch vegetation is strongly depredated and did not cover the all-farm perimeter. Therefore, the effect of landscape will be lower than we could expect. All farms studied are in the same oceanic-pluviseasonal Mediterranean bioclimate, with an upper meso-Mediterranean thermotype and a low subhumid ombrotype, as well as the same meso-Mediterranean Paeonio-Querco rotundifoliae sigmetum vegetation series. However, due to anthropozoogenic action, the natural replacement shrubs and phanerophyte remains around the farms studied are not always the same in all of them.

3) Two farms (Los Almendros and and Pinar left) has no pine in the surrounding (at least following the results in Herrera and Ruano, 2022). Since Pine support a lower population of Chrysopidae (again following the interesting results in Herrera and Ruano, 2022) this can have a great effect on Chrysopidae population.

Response: Alcalá Herrera and Ruano 2022 showed that pine support a lower population of Chrysopidae and a specific Chrysopidae species (Chrysoperla mediterranea) which has not significantly collected on almond, oak and olive trees. However oak tree was available in all farms and had the largest number of Chrysopidae adults collected per tree, as well as the highest species richness and diversity.

4) Tillage management has only one olive farm and (lines 367-368) in 2017 the sampling in this field has problem (only four transect instead of eight as in other fields). So we have only one field with critical data for one of the two sampling year: it is not correct to add this management in 2017 analysis.

Response: Thank you for your comments. We have deleted the analysis in 2017 to the row. We have modified the text in material and methods and results section (lines 155 to 194 and 453 to 455). We have established two-year sample to avoid the problem with the tillage management and to confirm our overall results to the vegetation cover management. In our results we have observed that the main families showed a similar pattern in both years.

In addition present research try to study the indirect effect of cover management on Chrysopidae though the effect of floristic composition in the olive grove. However no data are given of the intermediate state in this sequence; in particular no information is given about preys (both target or additional), other predators (with possible intraguild predaction effect), even if vacuum sampling allow to obtain these data. These information are important to understand if the supposed attraction of some plant is a per se effect or is due to change in other insect population.

Response: We are agreeing that vacuum sampling allow us to collect a wide range of arthropods such as parasitoids and phytophages. Other objective might be to determine the potential attraction of the plant species to other insect population as well as trophic interactions. However, our objective was to evaluate the potential attraction between plant families composition in the SNH (cover and patch vegetation) and the Chrysopidae community. Mostly of the Chrysopidae species collected in our study have a palyno-glycophagous diet. They feed on nectar, pollen and honeydew.  Previous studies have reported that some plant species are specifically associated with Chrysopidae species, some of which prefer trees species while others prefer herbaceous species (Alcalá Herrera et al., 2022, 2020; Monserrat and Marín, 2001, 1994; Villa et al., 2019; Villenave et al., 2013, 2006, 2005).  

Alcalá Herrera, R., Fernández Sierra, M.L., Ruano, F., Saunders, M.E., 2020. The suitability of native flowers as pollen sources for Chrysoperla lucasina (Neuroptera: Chrysopidae). PLoS ONE 15 (10), e0239847. http://sci-hub.tw/10.1371/journal.pone.0239847.

Alcalá Herrera, R., Cotes, B., Agustí, N., Tasin, M., Porcel, M., 2022. Using flower strips to promote green lacewings to control cabbage insect pests. J Pest Sci 95 (2), 669–683.

Monserrat, V.J., Marín, F., 2001. Comparative plant substrate specificity of Iberian Hemerobiidae, Coniopterygidae, and Chrysopidae, in: Whittington, A.E., McEwen, P. K., New, T.R. (Eds.), Lacewings in the Crop Environment. Cambridge University Press, Cambridge, UK, pp. 424–434. https://doi.org/DOI: 10.1017/ CBO9780511666117.026.

Monserrat, V.J., Marín, F., 1994. Plant substrate-specifity of Iberian Chrysopidae (Insecta, Neuroptera). Acta Oecologica-International J. Ecol. 15, 119–131.

Villa, M., Somavilla, I., Santos, S.A.P., López-Sáez, J.A., Pereira, J.A., 2019. Pollen feeding habits of Chrysoperla carnea s.l. adults in the olive grove agroecosystem. Agric. Ecosyst. Environ. 283, 106573 http://sci-hub.tw/10.1016/j.agee.2019.106573.

Villenave, J., Thierry, D., Al Mamun, A., Lode, T., Rat-Morris, E., 2005. The pollens consumed by common green lacewings Chrysoperla spp. (Neuroptera: Chrysopidae) in cabbage crop environment in western France. Eur. J. Entomol. 102, 547–552.

Villenave, J., Deutsch, B., Lode, T., Rat-Morris, E., 2006. Pollen preference of the Chrysoperla species (Neuroptera: Chrysopidae) occurring in the crop environment in western France. Eur. J. Entomol. 103, 771–777.

Villenave-Chasset, J., Denis, A., 2013. Study of pollen consumed by lacewings (Neuroptera, Chrysopidae) and hoverflies (Diptera, Syrphidae) in western France. Symbioses 29, 17–20.

I’m not a mother language, anyway I think that English check of the manuscript is necessary.

Response: Proofreading of the manuscript was done by Michael Lockwood. He is a native English speaker with more than 25 years of experience in proofreading research article.

Round 2

Reviewer 3 Report

In my previous review of this manuscript I underlined two problems.

1)     The choice to focus only on Chrysopidae

2)     The experimental design to evaluate the effect of cover management on plant vegetation and consequently of Chrysopidae population in olive orchards.

In both cases authors answer only marginally to the underlined points.

1)     I agree with authors about the importance of Chrysopidae as pest predators; my observation regard the choice to focus on Chrysopidae. I suppose that this choice has been done a priori and not following sampling (for example due to a higher abundance of Chrysopidae over other predators or parasitoids). In this case (a priori choice to focus on Chrysopidae) some general statement done in the manuscript are not correct. For example at the end of abstract (lines 38-40) authors stated that “According to our results, farmers may also increase the presence of other autochthonous plant species by seeding or planting in the cover and patch vegetation to increase the biological control in the agroecosystem”; this is not correct since we have no idea from the manuscript what happens to other predators/parasitoids which may be equally important in pest biological control in olive orchards. At the end of Introduction (Lines 109-111) I suggest to eliminate “natural enemy communities” since authors studied only Chrysopidae. In addition I agree with authors that in the Introduction they correctly underlined the importance of Chrysopidae as biological control agent in olive orchard, but they give no justification why they decide to not consider other beneficial groups. If the main objective of the research is to increase general biological control in olive orchard it is necessary to explain why they focus only on Chrysopidae and therefore why they did not consider other taxa. Therefore I think that to solve this point it is sufficient to eliminate some general statements.

2)     Second point is much more complicated. I think that authors have a panel of farms which is not adequate to solve the first objective of the manuscript (effect of different vegetation management on floristic composition in olive orchards). The number of fields is too low and additional differences in other variables (field dimension and plantation framework) are present other than the type of cover management. I agree with authors that cover management could is the main effect, but we have no data from the manuscript to substantiate this assumption. So if the main objective of the research is to study the effect of cover management on floristic composition (Lines 104-106) the set of orchards chosen is not adequate.

Author Response

Reviewer 3. Comments and Suggestions for Authors

In my previous review of this manuscript I underlined two problems.

  1. The choice to focus only on Chrysopidae
  2. The experimental design to evaluate the effect of cover management on plant vegetation and consequently of Chrysopidae population in olive orchards.

In both cases authors answer only marginally to the underlined points.

  1. I agree with authors about the importance of Chrysopidae as pest predators; my observation regard the choice to focus on Chrysopidae. I suppose that this choice has been done a priori and not following sampling (for example due to a higher abundance of Chrysopidae over other predators or parasitoids). In this case (a priori choice to focus on Chrysopidae) some general statement done in the manuscript are not correct. For example at the end of abstract (lines 38-40) authors stated that “According to our results, farmers may also increase the presence of other autochthonous plant species by seeding or planting in the cover and patch vegetation to increase the biological control in the agroecosystem”; this is not correct since we have no idea from the manuscript what happens to other predators/parasitoids which may be equally important in pest biological control in olive orchards. At the end of Introduction (Lines 109-111) I suggest to eliminate “natural enemy communities” since authors studied only Chrysopidae. In addition I agree with authors that in the Introduction they correctly underlined the importance of Chrysopidae as biological control agent in olive orchard, but they give no justification why they decide to not consider other beneficial groups. If the main objective of the research is to increase general biological control in olive orchard it is necessary to explain why they focus only on Chrysopidae and therefore why they did not consider other taxa. Therefore I think that to solve this point it is sufficient to eliminate some general statements.

Response: Thank you for your comments. We have modified the text according your comments (lines 38 to 40 and 113). We chose to focus on Chrysopidae due to their abundance and key role in the olive orchard. Previous studies showed that Chrysopidae plays a key role in the predation of the eggs and larvae of the olive moth, Prays oleae (Bernard, 1788), one of the principal olive pests (Alcalá Herrera and Ruano, 2022; Campos, 1989; González-Ruiz et al., 2006; Porcel et al., 2017; Ramos et al., 1984; Szentkirályi, 2001). Furthermore, the Integrated Pest Management guide recommends the use of Chrysopid as one biological control strategy against P. oleae infestation in olive orchard (Martín Gil and Ruíz Torres, 2014). We have underlined the importance of Chrysopidae as biological control agent in olive orchards following your suggestion (lines 64 to 74).

Alcalá Herrera, R., Ruano, F., 2022. Impact of woody semi-natural habitats on the abundance and diversity of green lacewings in olive orchards. Biol. Control. https://doi.org/10.1016/j.biocontrol.2022.105003

Campos, M., 1989. Observaciones sobre la bioecologia de Chrysoperla carnea (Stephens) (Neuroptera: Chrysopidae) en el sur de España. Neuroptera Int. 5, 159–164.

González-Ruiz, R., Al-Asaad, S., Bozsik, A., 2006. The influence of the adjacent vegetation patches on diversity and abundance of green lacewings associated to the olive groves in south Spain. Implications in the Natural control of the olive moth, Prays oleae (Lep: Yponomeutidae), in: Symposium at Debrecen University.

Martín Gil, A., Ruíz Torres, M.J., 2014. Guía de gestión integrada de plagas olivar. Ministerio de Agricultura, Alimentación y Medio Ambiente, Gobierno de España, Madrid, ES.

Porcel, M., Cotes, B., Castro, J., Campos, M., 2017. The effect of resident vegetation cover on abundance and diversity of green lacewings (Neuroptera: Chrysopidae) on olive trees. J. Pest Sci. (2004). 90, 195–196. https://doi.org/10.1007/s10340-016-0748-5

Ramos, P., Campos, M., Ramos, J.M., 1984. Estabilización del ataque de Prays oleae Bern. y de la actividad de los depredadores oófagos sobre el fruto del olivo. Boletín Sanid. Veg. Plagas 10, 239–243.

Szentkirályi, F., 2001. Lacewings in fruit and nut crops, in: Whittington, A.E., McEwen, P.K., New, T.R. (Eds.), Lacewings in the Crop Environment. Cambridge University Press, Cambridge, UK, pp. 172–238. https://doi.org/DOI: 10.1017/CBO9780511666117.011

  1. Second point is much more complicated. I think that authors have a panel of farms which is not adequate to solve the first objective of the manuscript (effect of different vegetation management on floristic composition in olive orchards). The number of fields is too low and additional differences in other variables (field dimension and plantation framework) are present other than the type of cover management. I agree with authors that cover management could is the main effect, but we have no data from the manuscript to substantiate this assumption. So if the main objective of the research is to study the effect of cover management on floristic composition (Lines 104-106) the set of orchards chosen is not adequate.

Response: We are agreeing with the reviewer that larger samples or farms would provide more statistical certainty, we have tried to solve this issue sampling two consecutive years in each farm. All transects could be done correctly regardless of the surface farm under the same methodology, which is specifically designed to herbaceous species and our study scale. We carried out sixteen cover vegetation transects in total in both years per farm, except to Los Almendros with 12 transects and we have justified why, as well as the text have been modified according your comments in the previous round. Each transect was determined by contact at 100 points, five centimetres apart, along a 5 m, with a minimum distance of 25 m between transects to keep the independence between transect. This sampling effort attempts to compensate for the impossibility of increasing the number of sampled farms, due to the lack of availability of more close farms under the same management and other abiotic variables.

Regarding to the plantation framework, the difference among our farms is four meters. We sincerely believe that the cover vegetation has enough space to emerge properly in the middle of the row and beneath the tree canopy and the plantation framework has not an effect in our study. I would have agreed with your statement about the plantation framework, if the olive orchard framework had been a mixed between an extreme traditional (14 x 14 m), intensive (7 x 7 m or less) or trellis (1.35 to 1.5 x 3.5 to 4 m) framework, as a recent study has showed (Vasconcelos et al., 2022), but not in our case. All farm selected belonged to the same vegetation series (Paeonio-Querco rotundifoliae sigmetum), pesticide management, soil traits, olive variety, and weather conditions (an oceanic-pluviseasonal Mediterranean bioclimate, with an upper meso-Mediterranean thermotype and a low subhumid ombrotype), as well as similar altitude and traditional framework (between 12 x 12 or 8 x 8 m). Furthermore, the cover vegetation and the cover vegetation management in each farm have at least a minimum of four-years, which are old enough to avoid the floristic changes due to the succession after a possible cover vegetation restoration. In fact, farms with a long-term grazing cover management has larger number of leguminous and composite plant species, as well as the high abundance of plant species with a basal rosette shape, determine that the cover vegetation is stable.

Regarding to the patch vegetation, it is typical from the mentioned vegetation series as well as conditioned by the historical and anthropic activities in the whole region, in which most of the trees have been felled. Nevertheless, remains of the serial stages, which are maintained for decades, only could be affected by recurrent fires in the Mediterranean region. Fire events have not happened in the chosen farms during our study period.

For all that reasons, we consider that we have enough data and the farms selected are correct to support our objective “determine the effect of the cover management techniques (grazing, mowing and tillage) on the floristic composition in the cover vegetation in organic olive farms”. We have described our study area in lines 327 to 367.

Vasconcelos, S., Pina, S., Herrera, J.M., Silva, B., Sousa, P., Porto, M., Melguizo-Ruiz, N., Jiménez-Navarro, G., Ferreira, S., Moreira, F., Jonsson, M., Beja, P., 2022. Canopy arthropod declines along a gradient of olive farming intensification. Sci. Rep. 12. https://doi.org/10.1038/s41598-022-21480-1
